# Osteoprotegerin genetic polymorphisms and their influence on therapeutic response to ibandronate in postmenopausal osteoporotic females

Saba Tariq[1,2]*, Sundus Tariq[3], Shahad Abduljalil Abualhamael[4], Mukhtiar Baig[5], Ahmad Azam Malik[6], Muhammad Shahzad[7]

1 Department of Pharmacology and Therapeutics, University Medical & Dental College, The University of Faisalabad, Faisalabad, Pakistan, 2 University of Birmingham, Birmingham, England, United Kingdom, 3 Department of Physiology, International School of Medicine, Istanbul Medipol University, Research Institute for Health Sciences and Technologies (SABITA), Istanbul, Turkey, 4 Department of Internal Medicine, Faculty of Medicine in Rabigh, King Abdulaziz University, Jeddah, Saudi Arabia, 5 Department of Clinical Biochemistry, Faculty of Medicine in Rabigh, King Abdulaziz University, Jeddah, Saudi Arabia, 6 Department of Family and Community Medicine, Faculty of Medicine in Rabigh, King Abdulaziz University, Jeddah, Saudi Arabia, 7 Department of Pharmacology, University of Health Sciences, Lahore, Pakistan

* drsabatariq1@gmail.com, s.tariq@bham.ac.uk

**Data Availability Statement:** All relevant data are within the paper and its Supporting Information files.

## Abstract

### Objectives

The present study investigated osteoprotegerin (OPG) genetic polymorphisms and their influence on the therapeutic response to ibandronate in postmenopausal osteoporotic females.

### Methods

This case-control study included 135 postmenopausal females (89 osteoporotic females and 46 non-osteoporotic females). Each osteoporotic patient received a monthly 150 mg ibandronate tablet for six months, and blood samples were taken before and after treatment. Bone mineral density (BMD) was measured using DEXA Scan. Three SNPs (A163G, T245G, and G1181C) of the OPG gene were selected for analysis.

### Results

Serum OPG levels were significantly lower in osteoporotic subjects than in the control group. The percentage changes in OPG levels in the osteoporotic group before and after treatment with ibandronate were significant (p < .001). After six months of therapy with ibandronate, the percentage changes in OPG levels with AA, TT, TC, GC, and GG genotypes were significant. Following six months of ibandronate treatment, the AA genotype of rs3134069, TT, TC genotypes of rs3102735, GG, and GC genotypes of rs2073618 SNP showed a significant increase in OPG levels. Age, BMI, and GC polymorphism (rs2073618 (G/C) G1181C) were inversely associated with low BMD. Adjusted odds ratios (OR) showed

**Funding:** The author(s) received no specific funding for this work.

**Competing interests:** The authors have declared that no competing interests exist.

that BMI, GC, GG polymorphism (rs2073618 (G/C) G1181C) and TC polymorphism (rs3102735 (T/C) A163G) were inversely associated with low BMD.

## Conclusion

The inverse association of rs2073618 and rs3102735 with low BMD indicates the protective role of these SNPs in our population. More research is needed to replicate these results in another cohort and to determine the molecular processes by which such SNPs may influence BMD.

## Introduction

Bone remodeling is one of the homeostasis mechanisms the human body encounters throughout life in the skeletal system. One of the major pathways involved in bone homeostasis includes the RANKL/RANK/OPG system, which produces signals maintaining the balance between bone resorption and formation, indicating that this pathway has a role in diagnostic and therapeutic interventions in bone diseases such as osteoporosis [1]. Osteoblasts release a protein called osteoprotegerin (OPG) that can attach to the RANK ligand (RANKL) and act as RANKL's natural inhibitor. This process is balanced in premenopausal women; however, in postmenopausal women, a drop in estrogen causes an increase in RANKL expression, which bypasses OPG and results in augmented binding with RANK, increasing osteoclast activity and bone resorption, which ultimately results in osteoporosis [2].

Different studies have documented that single nucleotide polymorphisms (SNPs) in TNFRSF11B (OPG) genes are correlated with a decrease in BMD and fracture risk [3, 4]. The OPG gene is believed to engage in the pathogenesis of osteoporosis [5]. In postmenopausal women, the SNP A163G in the promoter region of this gene is linked to lower bone mineral density BMD [6]. Another study suggested that the OPG/A163G polymorphism participates in the genetic control of bone homeostasis among the Slovak population and can raise or lower the risk of osteoporosis in such individuals [7]. Different meta-analyses (MA) have shown that the G allele of the OPG A163G and T245G polymorphisms might increase osteoporosis risk [8, 9]. One interesting study displayed that A163G and T245G were linked to augmented fracture risk. In contrast, people with the C-allele of the G1181C polymorphism may have a lowered osteoporotic risk, particularly among women of Asian origin and postmenopausal females [10]. A recent study reported that genetic polymorphisms in the *VDR* gene may affect the effectiveness of ibandronate and raloxifene treatment osteoporotic females [11]. Similarly, several types of polymorphism in OPG gene are associated with fracture risk in osteoporotic female patients [8, 9]. A number of studies have also reported the age and body mass index BMI relationship with BMD [12–14].

Bisphosphonates are powerful drugs preventing bone resorption and often cure metabolic bone diseases. The accurate pathway by which bisphosphonates perform their action is still unclear. Therefore, many studies are being conducted to determine the accurate regulatory effect of bisphosphonate [15]. To date, there is a lack of information evaluating the relationship between the OPG gene and BMD in our local population. Moreover, no published data describing the OPG gene SNP and its association with osteoporosis in the Pakistani population is available. Therefore, the present study was designed to identify OPG genetic polymorphisms and their influence on the therapeutic response to ibandronate in Pakistani postmenopausal osteoporotic females.

## Methods

### Study design and setting

For this cross-sectional study, 135 postmenopausal women were chosen from the Madina teaching hospital, a tertiary care hospital in Faisalabad, Pakistan. Ethical approval was obtained from the Institutional Review Board of the University of Health Sciences, Lahore. The protocol for data collection adhered to institutional, national, and Helsinki Declaration ethical norms. Data confidentiality and anonymity were maintained.

### Data collection and inclusion and exclusion criteria

Initially, these females underwent BMD assessment using a calcaneal ultrasound scan. Additionally, BMD at the lumbar spine (L2-L4), right femoral neck, right hip, left femoral neck, and left hip was evaluated through dual-energy X-ray absorptiometry (DEXA) using HOLO-GIC-HORIZON (QDR-series) dual-energy X-ray absorptiometry system to confirm BMD. DXA results were utilized for final analysis and presented as a T-score, comparing the measured BMD with the average BMD of young adults at peak bone mass. BMD was assessed at the study's inception, but not after six months, as literature indicates minimal changes in BMD occur within the first 12 months [16].

Females were categorized into two groups. The osteoporotic group comprised females aged 50 to 70, postmenopausal for over a year, and with T scores less than or equal to -2.5. All females with liver, renal or gastrointestinal diseases were excluded, as were females on medication, such as bisphosphonates, hormonal replacement therapy, corticosteroids, or those with any other bone or metabolic diseases. The control group consisted of 46 postmenopausal non-osteoporotic healthy females aged 50 to 70 with menopause for over a year and with T scores greater than or equal to -1.0. With 90% power of the study and 95% confidence level, 37 sample size was determined using the following formula.

$$n = 2\ S^2 (Z_{\alpha} + Z_{\beta})^2 /\ d^2$$

All participants provided written informed consent. Their demographic data, including age, height, weight, and BMI, were collected on a specifically designed proforma. Blood samples were taken at the beginning of the research and again after six months of ibandronate medication. Each osteoporotic patient received one bisphosphonate pill (ibandronate 150 mg) per month for six months. Serum OPG levels were measured using ELISA kits, and its absorbance was measured at a wavelength of 450 nm. The OD of the samples was compared to the standard curve for OPG concentration calculation Elab Science Biotechnology Incorporation supplied the OPG kit. Biochemical analysis used microplate data collection and analysis software Gen5TM and Gen5 Secure by BioTekVR Instruments, Inc. DNA was extracted from whole blood using GeneJet Whole Blood Genomic DNA Purification Mini Kit by Thermo Fisher Scientific Inc., Carlsbad, California 92008, USA. The samples were sent to Advance Bioscience International (ABI) China for sequencing. The data obtained after sequencing was then visualized using Bio Edit Software and then analyzed using BLAST (NCBI) to find out the variations in these sequences. Already reported primers were chosen and rechecked with sequences from the NCBI website, while the longer sequence was obtained from the UCSC website (http://genome.ucsc.edu/). Primers' specificity was validated using the NCBI database of the human genome with BLAST (http://www.ncbi.nlm.nih.gov/BLAST/). We selected three OPG gene SNPs: A163G, T245G, and G1181C.

**A163G (rs3102735).**   This SNP is present in the promoter region (5′UTR) of the TNFRSF11B (OPG) gene for analysis; such polymorphism is associated with osteoporosis and other bone-related conditions [6].

**T245G (rs3134069).**   This SNP is present in the promoter region (5′UTR) of the TNFRSF11B (OPG) gene for analysis; such polymorphism is associated with osteoporosis and other bone-related conditions [9, 17].

**G1181C (rs2073618).**   This SNP is located in the first exon of the TNFRSF11B (OPG) gene for analysis; such polymorphism is associated with osteoporosis and other bone-related conditions [9, 17].

Genomic DNA was extracted from whole blood samples using a DNA kit. Samples were stored at -70˚C. Sequencing was carried out by Advance Bioscience International (ABI) China. Sequencing data were visualized using Bio Edit Tools Software and analyzed with BLAST (NCBI) to identify sequence differences.

## Statistical analysis

Statistical analysis was performed using SPSS 24. Genotype frequencies of cases and controls were calculated. Student's t-test was applied to compare age, height, weight, BMI, OPG, and BMD between osteoporotic patients and healthy controls.

Mann–Whitney U test and Kruskal–Wallis test were used to compare bone mineral density between different genotypes. Mann–Whitney U test and Kruskal–Wallis test were also employed to compare percentage changes in OPG concentrations after six months of ibandronate treatment in the osteoporotic group among different genotypes. Logistic regression analysis was employed to evaluate age, BMI, rs3134069 (A/C) T245G, rs3102735 (T/C) A163G, and rs2073618 (G/C) G1181C polymorphisms as predictors of BMD. Odds ratios (OR) and adjusted OR were calculated. Statistical significance was set at a p-value less than 0.05.

## Results

### Participants' characteristics

A total of 135 females participated in this study, with 89 osteoporotic and 46 healthy non-osteoporotic females. Serum OPG and BMD levels in osteoporotic individual' were significantly lower than those in the control group. A comparison of other characteristics, such as age, height, weight, and BMI, between osteoporotic patients and healthy controls is presented in Table 1. Significant percentage changes in OPG levels were observed in the osteoporotic group before and after ibandronate treatment (p < .001) (Fig 1).

### Genotype and allele frequencies for OPG genetic polymorphisms

Table 2 displays the genotype and allele frequencies for OPG genetic polymorphisms (SNP1: rs3134069, SNP2: rs3102735, SNP3: rs2073618). For rs3134069, the genotypic frequencies in osteoporotic patients (AA: 88%, AC: 12%, CC: 0%) were not statistically different from those in healthy controls (AA: 86%, AC: 14%, CC: 0%; $\chi^2$ = 0.63, p = 0.42). Similarly, no significant differences were found in allele frequencies between osteoporotic patients (A: 94%, C:11%) and healthy controls (A: 91%, C: 9%; $\chi^2$ = 0.58, p = 0.44). Likewise, no significant differences in genotype and allele frequencies were identified for OPG genetic polymorphisms rs3102735 and rs2073618 (Table 2).

**Table 1. Comparisons of age, height, weight, BMI, osteoprotegerin, and BMD between osteoporotic patients and healthy controls.**

| Variables | Group | N | Mean | SD | ^p-value |
|---|---|---|---|---|---|
| Age (years) | Normal | 46 | 60.5 | 5.6 | 0.040* |
| | Osteoporotic | 89 | 58.3 | 6.0 | |
| Height (meters) | Normal | 46 | 1.6 | 0.0 | 0.009* |
| | Osteoporotic | 89 | 1.5 | 0.1 | |
| Weight (Kg) | Normal | 46 | 75.4 | 12.3 | < 0.001* |
| | Osteoporotic | 89 | 62.3 | 13.3 | |
| BMI (Kg/m$^2$) | Normal | 46 | 31.1 | 4.8 | < 0.001* |
| | Osteoporotic | 89 | 26.4 | 5.3 | |
| Osteoprotegerin (ng/ml) | Normal | 46 | 13.33 | 2.94 | < 0.001* |
| | Osteoporotic | 89 | 11.34 | 2.77 | |
| Lumbar spine BMD, mean ± SD | Normal | 46 | 0.05 | 1.16 | < 0.001* |
| | Osteoporotic | 89 | -2.88 | 0.86 | |
| Right Femoral Neck BMD, mean ± SD | Normal | 46 | -0.15 | 1.24 | < 0.001* |
| | Osteoporotic | 89 | -2.12 | 1.02 | |
| Right Hip BMD, mean ± SD | Normal | 46 | 0.19 | 0.93 | < 0.001* |
| | Osteoporotic | 89 | -1.69 | 1.09 | |

^Mann–Whitney U test was employed

*p-value <0.05 was considered significant

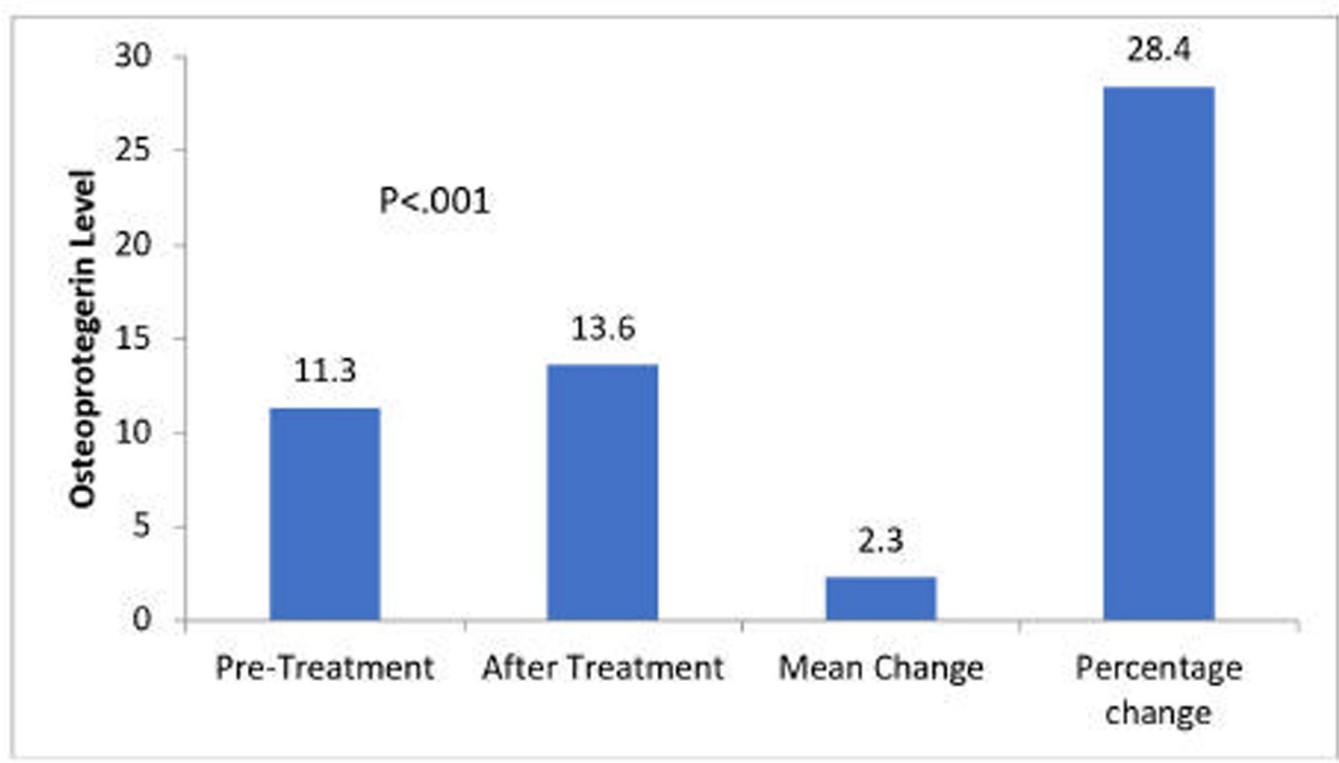

**Fig 1. Changes in osteoprotegerin levels after six months of treatment with ibandronate.**

**Table 2. Comparison of genotypic and allelic frequencies of OPG genetic polymorphism in osteoporotic and control groups.**

| Groups | SNP1: rs3134069 | | | | | SNP2: rs3102735 | | | | | SNP3: rs2073618 | | | | |
|---|---|---|---|---|---|---|---|---|---|---|---|---|---|---|---|
| | Genotype frequencies (%) | | | Allelic frequencies (%) | | Genotype frequencies (%) | | | Allelic frequencies (%) | | Genotype frequencies (%) | | | Allelic frequencies (%) | |
| | AA | AC | CC | A | C | TT | TC | CC | T | C | CC | CG | GG | C | G |
| Osteoporotic (n =) | 78 (88) | 11 (12) | 0 | 167 (94) | 11 (6) | 58 (65) | 29 (33) | 2 (2) | 145 (81) | 33 (19) | 34 (38) | 42 (47) | 13 (15) | 110 (62) | 68 (38) |
| Control (n =) | 38 (83) | 8 (17) | 0 | 84 (91) | 8 (9) | 36 (78) | 10 (22) | 0 (0) | 82 (89) | 10 (11) | 10 (22) | 29 (63) | 7 (15) | 49 (53) | 43 (47) |
| Total(n =) | 116 (86) | 19 (14) | 0 (0) | 251 (93) | 19 (7) | 94 (7) | 39 (29) | 2 (1) | 227 (84) | 43 (43) | 44 (33) | 71 (53) | 20 (15) | 159 (59) | 111 (41) |
| Chi-square | $X^2 = 0.63$ | | | $X^2 = 0.58$ | | $X^2 = 1.95$ | | | $X^2 = 2.66$ | | $X^2 = 3.97$ | | | $X^2 = 1.82$ | |
| *P-value | P = .42 | | | P = .44 | | P = .16 | | | P = .10 | | P = .13 | | | P = .17 | |

*Chi-square test was employed

### Comparison of BMD between different genotypes

In genotype rs3134069 (A/C) T245G, there was no significant variations in BMD among osteoporotic patients at any site. However, BMD significantly differed in healthy controls at all sites except the lumbar region. In genotype rs3102735 (T/C) A163G, no significant differences were observed in BMD in osteoporotic patients at any site. However, BMD significantly differed in healthy controls at all sites except the lumbar and left hip regions. For rs2073618 (G/C) G1181C, no significant differences were observed in BMD among osteoporotic patients and the control group at any site (Table 3).

### Comparison of percentage changes in OPG levels after treatment with ibandronate in the osteoporotic group among different genotypes

After six months of ibandronate therapy, the percentage variations in OPG concentrations in the osteoporotic group with AA, TT, TC, GC, and GG genotypes were significant (Table 4).

### Predictors of low bone mineral density

Logistic regression analysis evaluated age, BMI, rs3134069 (A/C) T245G, rs3102735 (T/C) A163G, and rs2073618 (G/C) G1181C polymorphisms as predictors of low BMD. The osteoporotic group was considered the low BMD group. Age, BMI, and the GC polymorphism (rs2073618 (G/C) G1181C) were inversely related to low BMD. Adjusted OR indicated that BMI, GC, and GG polymorphism (rs2073618 (G/C) G1181C) and TC polymorphism (rs3102735 (T/C) A163G) were inversely associated with low BMD (Table 5).

## Discussion

A review of the literature revealed a paucity of evidence linking ibandronate treatment response to OPG genetic variants in postmenopausal osteoporotic females. However, a few investigations have indicated that bisphosphonate increases OPG gene expression and protein production in human osteoblasts in a dose-dependent manner [15, 18]. Additionally, they also reported that the effect of bisphosphonates on osteoblastic OPG protein secretion grew sixfold over time [15, 18].

The significant increase in OPG levels after 6 months of ibandronate medication in the osteoporotic group was seen in SNPs rs3134069 (A/C)T245G (AA genotype), SNPs rs3102735 (T/C) A163G, (TT and TC genotypes), and rs2073618 G1181C (GG and GC) genotypes, indicating that people with these genotype variants responded effectively to treatment in terms of increasing OPG levels. Villagómez et al. (2023) showed that OPG SNPs rs2073618 and

**Table 3. Comparison of bone mineral density between different genotypes.**

| Group | rs3134069 (A/C)T245G | N | Lumbar | Right Femoral Neck | Right Hip | Left Femoral Neck | Left Hip |
|---|---|---|---|---|---|---|---|
| Normal | AA | 39 | -0.12 ± 1.07 | 0.12 ± 1.10 | 0.31 ± 0.94 | 0.09 ± 1.11 | 0.41 ± 0.93 |
| | AC | 07 | 1.04 ± 1.21 | -1.63 ± 0.88 | -0.47 ± 0.52 | -1.00 ± 0.67 | -0.24 ± 0.68 |
| | p-value | | 0.052 | 0.001* | 0.019* | 0.013* | 0.045* |
| Osteoporotic | AA | 78 | -2.86 ± 0.90 | -2.10 ± 1.04 | -1.66 ± 1.09 | -1.93 ± 1.00 | -1.48 ± 1.09 |
| | AC | 11 | -3.04 ± 0.49 | -2.26 ± 0.92 | -1.88 ± 1.08 | -2.18 ± 0.83 | -1.67 ± 0.57 |
| | p-value | | 0.352 | 0.557 | 0.600 | 0.541 | 0.881 |
| Group | rs3102735 (T/C) A163G | N | Lumbar | Right Femoral Neck | Right Hip | Left Femoral Neck | Left Hip |
| Normal | TT | 36 | -0.08 ± 1.10 | 0.12 ± 1.13 | 0.32 ± 0.96 | 0.10 ± 1.15 | 0.43 ± 0.95 |
| | TC | 10 | 0.52 ± 1.31 | -1.12 ± 1.16 | -0.25 ± 0.68 | -0.71 ± 0.77 | -0.12 ± 0.64 |
| | p-value [a] | | 0.627 | 0.007* | 0.046* | 0.049* | 0.053 |
| Osteoporotic | TT | 58 | -2.86 ± 0.92 | -2.13 ± 1.08 | -1.66 ± 1.08 | -1.93 ± 1.03 | -1.42 ± 1.02 |
| | TC | 29 | -2.87 ± 0.73 | -2.11 ± 0.95 | -1.73 ± 1.16 | -2.06 ± 0.87 | -1.67 ± 1.12 |
| | CC | 2 | -3.60 ± 0.85 | -2.15 ± 0.78 | -2.00 ± 0.57 | -1.65 ± 1.48 | -1.40 ± 0.57 |
| | p-value [b] | | 0.377 | 0.998 | 0.776 | 0.962 | 0.489 |
| Group | rs2073618 (G/C) G1181C | N | Lumbar | Right Femoral Neck | Right Hip | Left Femoral Neck | Left Hip |
| Normal | CC | 10 | -0.04 ± 1.15 | -0.18 ± 0.52 | 0.11 ± 0.65 | -0.55 ± 0.63 | 0.03 ± 0.62 |
| | GC | 29 | 0.003 ± 1.25 | -0.03 ± 1.41 | 0.19 ± 0.98 | 0.08 ± 1.31 | 0.36 ± 1.01 |
| | GG | 7 | 0.43 ± 0.74 | -0.61 ± 1.23 | 0.33 ± 1.15 | -0.04 ± 0.57 | 0.47 ± 0.89 |
| | p-value | | 0.491 | 0.409 | 0.896 | 0.312 | 0.548 |
| Osteoporotic | CC | 34 | -2.87 ± 0.75 | -2.18 ± 1.04 | -1.69 ± 1.10 | -1.92 ± 0.92 | -1.43 ± 0.96 |
| | GC | 42 | -2.86 ± 0.97 | -2.09 ± 1.03 | -1.66 ± 1.06 | -2.00 ± 1.06 | -1.49 ± 1.09 |
| | GG | 13 | -2.97 ± 0.76 | -2.07 ± 1.00 | -1.78 ± 1.22 | -1.95 ± 0.95 | -1.72 ± 1.13 |
| | p-value | | 0.891 | 0.969 | 0.786 | 0.727 | 0.642 |

Mann–Whitney U test, Kruskal–Wallis test

rs3102735 of the OPG gene responded to bisphosphonate (alendronate) treatment (68%), while 32% of postmenopausal osteoporotic females had a poor response to bisphosphonate (alendronate) due to gene variants [19]. Another study demonstrated that osteoporosis

**Table 4. Comparison of percentage changes in osteoprotegerin levels after treatment with ibandronate in the osteoporotic group (median (IQR) among different genotypes.**

| Geno Types | N | Pre-Treatment | After Treatment | Median Change | Percentage change | Wilcoxon signed-rank test p-value |
|---|---|---|---|---|---|---|
| **rs3134069 (A/C) T245G** | | | | | | |
| **AA** | 78 | 11.3 (9.5–13.0) | 13.0 (10.4–16.7) | 1.95 (-0.97–3.7) | 18.0 (-7.3–39.8) | 0.001* |
| **AC** | 11 | 11.3 (9.9–11.9) | 13.0 (7.8–17.5) | 1.84 (-2.4–7.5) | 15.5 (-21.7–65.6) | 0.285 |
| Mann–Whitney U test p-value | | 0.579 | 0.699 | 0.921 | 0.990 | |
| **rs3102735 (T/C) A163G** | | | | | | |
| TT | 58 | 11.3 (9.5–12.7) | 12.0 (9.8–14.9) | 1.85 (-1.5–3.7) | 16.1 (-12.1–32.9) | 0.007* |
| TC | 29 | 11.4 (9.9–13.8) | 15.1 (11.0–18.3) | 3.1 (-2.1–7.8) | 22.6 (-16.3–70.5) | 0.027* |
| CC | 2 | 10.8 (8.7 –N.A) | 12.8 (12.6 –N.A) | 1.97 (0.0 –N.A) | 22.8 (0.0 –N.A) | 0.317 |
| Kruskal–Wallis test p-value | | 0.842 | 0.241 | 0.698 | 0.795 | |
| **rs2073618 (G/C) G1181C** | | | | | | |
| CC | 34 | 11.5 (10.0–13.0) | 12.0 (8.8–15.6) | 1.14 (-2.9–4.1) | 8.0 (-24.9–39.7) | 0.174 |
| GC | 42 | 11.3 (9.2–12.6) | 13.6 (10.7–15.6) | 1.91 (-0.49–3.8) | 20.1 (-3.85–41.0) | 0.003* |
| GG | 13 | 9.3 (10.3–12.9) | 14.5 (10.0–15.6) | 3.62 (-0.91–8.43) | 27.2 (-9.0–80.0) | 0.039* |
| Kruskal–Wallis test p-value | | 0.563 | 0.469 | 0.296 | 0.369 | |

**Table 5. Logistic regression analysis to evaluate age, BMI, rs3134069 (A/C) T245G, rs3102735 (T/C) A163G and rs2073618 (G/C) G1181C polymorphism as a predictor of low bone mineral density (BMD).**

| Variables | Categories | β | p-value | OR | 95% CI | | p-value | aOR | 95% CI | |
|---|---|---|---|---|---|---|---|---|---|---|
| | | | | | Lower | Upper | | | Lower | Upper |
| Age | continuous | -0.064 | 0.041* | 0.938 | 0.881 | 0.998 | 0.117 | 0.945 | 0.880 | 1.014 |
| BMI | continuous | -0.172 | 0.000* | 0.842 | 0.779 | 0.911 | < 0.001* | 0.848 | 0.780 | 0.922 |
| rs3134069 (A/C)T245G | AA | Reference | | | | | | | | |
| | AC | -0.241 | 0.644 | 0.786 | 0.283 | 2.185 | 0.115 | 0.262 | 0.050 | 1.383 |
| rs3102735 (T/C) A163G | TT | Reference | | | | | | | | |
| | TC | 0.588 | 0.165 | 1.800 | 0.785 | 4.130 | 0.011* | 5.436 | 1.469 | 20.11 |
| | CC | - | - | - | - | - | - | - | - | - |
| rs2073618 (G/C) G1181C | CC | Reference | | | | | | | | |
| | GC | -0.853 | 0.049* | 0.426 | 0.182 | 0.996 | 0.006* | 0.253 | 0.096 | 0.672 |
| | GG | -0.605 | 0.306 | 0.546 | 0.172 | 1.739 | 0.036* | 0.229 | 0.058 | 0.908 |

OR = odds ratio

aOR = adjusted odds ratio

* = Significant

— = no case reported in control group

patients who did not respond to bisphosphonate treatment (40%) had a higher frequency of gene variations than patients who did respond to this medicine [20].

The bone environment is complex, and OPG is a powerful inhibitor of osteoclast formation; thus, ibandronate, by increasing the levels of OPG, helps slow the progression of the disease [21]. A meta-analysis of the OPG gene (TNFSRB11B) concluded that three polymorphisms identified as rs2073618, rs3134069, and rs3134070 had a protective outcome and reduced the chance of fracture [22]. Ibandronate also reduces the fracture risk in individuals with osteoporosis [23, 24]. The increased levels of OPG after treatment with ibandronate and the presence of these polymorphisms enhance the efficacy of ibandronate in such patients.

The odds ratio revealed that age, BMI, and GC polymorphism (rs2073618 (G/C) G1181C) were inversely related to low BMD, whereas adjusted OR indicated that BMI, GC, and GG polymorphism (rs2073618 (G/C) G1181C) and TC polymorphism (rs3102735 (T/C) A163G) were inversely related to low BMD. Our findings are consistent with a few earlier studies that discovered a substantial connection between SNPs and BMD in other groups [25–28]. In contrast to our findings, many studies found no association between serum OPG and BMD [29–31]. Similar to the findings of the present study, an Indian investigation found that SNPs rs2073618 and rs3102735 in the OPG gene may affect healthy women's BMD at the spine [32].

A Russian study reported that polymorphic OPG rs3134069, rs3102734, rs7844539, and rs3102734 are prospective risk markers for osteoporotic fractures and low BMD in men and women population. They found no link between rs3102735, rs2073618 of the OPG gene and osteoporotic fractures risk and the BMD level [4]. A recent MA indicated a link between the T245G polymorphism and osteoporosis risk. Furthermore, women with the GG or CG genotypes at the G1181C gene had a lower incidence of osteoporosis. Only the GG/GA genotypes at the A163G variation were more prone to developing osteoporosis, although CC/CG carriers of the G1181C locus may have a lower risk. The study suggested that these genetic markers could be employed as osteoporosis prediction tools [3]. A recent MA reported the association of OPG T950C polymorphism with the risk of osteoporosis among postmenopausal Chinese women [33].

The debate on BMI as a predictor of BMD is ongoing. BMI can predict BMD due to the common stromal cell origin of adipocytes and osteoblasts. A significant relationship between BMI and BMD among postmenopausal women was found in a few studies [14, 34].

A study stated that the possible link between BMI and BMD could be due to bone tissue and fat tissue originating from common stromal cells [35]. Our results are consistent with another study in which increased BMI was associated with a decreased risk of osteoporosis and increased BMD, especially at the femoral neck [36]. Recently, a study concluded that sedentary women with low BMI had significantly lower BMD [37]. However, contrasting results were also observed, as few studies showed that increased weight and BMI could increase the fracture rate and low BMD [38, 39]. This difference could be because although a higher BMI may result in greater BMD due to more immense pressures on the skeleton, it cannot determine fat mass and distribution. The pattern of obesity and fat distribution, on the other hand, may impact the rate of osteoporosis. Therefore, the fat distribution pattern might cause BMI's deleterious effects on bone density.

The present study found an association between age with BMD. Similarly, few other studies reported that an increase in age is linked to a drop in BMD due to the decline in estrogen in postmenopausal females [40]. In contrast, a study observed no association of age with BMD at the femoral neck [41], and other Pakistani studies have reported a negative correlation between age and BMD [13, 40].

Osteoporosis is a prevalent condition, particularly among postmenopausal women. The current study adds to the body of evidence pointing to the function of OPG gene polymorphisms in BMD. More research is needed to replicate these results in another cohort and to determine the molecular processes by which such SNPs may influence BMD.

## Strengths of the study

This study has many strengths. It explores an area by investigating how genetic variations in OPG impact the response to ibandronate in postmenopausal women with osteoporosis, providing new insights. The study has a substantial and diverse sample size of 135 individuals, ensuring power and generalizability. The researchers maintain rigor by conducting assessments of bone health using DEXA scans and measuring serum OPG levels through ELISA kits. Additionally, it investigates the influence of three gene variations in OPG, offering a genetic perspective on treatment outcomes. The study's direct evaluation of how ibandronate affects bone health by measuring percentage changes in OPG levels after six months of treatment highlights its clinical relevance. By considering factors like age, BMI, and genetics, the study provides an understanding of the variables that impact bone health.

## Limitations

The main weakness of our study was the limited sample size, unequal number of cases and controls, and we only did DEXA scanning on the patients once during the trial at the start of the treatment. Our study evaluated the influence of OPG genetic variants on the ibandronate therapeutic response after only six months of treatment; it is possible that a longer duration might be required. Longitudinal investigations are needed to assess the long-term impact of OPG genetic variants on ibandronate treatment response and analyze the inheritance pattern of osteoporotic genes in the Pakistani population.

## Conclusion

Following six months of treatment, we observed a significant increase in OPG levels in specific genotypes, including the AA genotype of rs3134069 SNP, as well as the TT and TC genotypes

of rs3102735, and the GG and GC genotypes of rs2073618 SNP. Moreover, our investigation uncovered an intriguing inverse relationship between the GC and GG polymorphisms of rs2073618 (G/C) G1181C and the TC polymorphism of rs3102735 (T/C) A163G, respectively, with BMD. These findings suggest that these SNPs may be protective in our community, potentially influencing bone health outcomes. While these results are promising, it is essential to interpret them cautiously and recognize the need for further research to validate and expand upon these associations.

## Supporting information

**S1 Data.**
(XLSX)

## Acknowledgments

This study abstract has been accepted and published in the proceedings of the World Congress on Osteoporosis, Osteoarthritis, and Musculoskeletal Diseases (WCO-IOF-ESCEO 2022).

## Author Contributions

**Conceptualization:** Saba Tariq, Muhammad Shahzad.

**Data curation:** Saba Tariq, Sundus Tariq, Shahad Abduljalil Abualhamael, Mukhtiar Baig.

**Formal analysis:** Mukhtiar Baig, Ahmad Azam Malik.

**Investigation:** Saba Tariq, Muhammad Shahzad.

**Methodology:** Saba Tariq, Sundus Tariq, Ahmad Azam Malik.

**Project administration:** Sundus Tariq, Muhammad Shahzad.

**Resources:** Sundus Tariq.

**Supervision:** Muhammad Shahzad.

**Validation:** Shahad Abduljalil Abualhamael.

**Writing – original draft:** Shahad Abduljalil Abualhamael, Ahmad Azam Malik.

**Writing – review & editing:** Mukhtiar Baig.

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
