## [Decision Letter · Decision Letter 0]

2 Aug 2023

PONE-D-22-35434Osteoprotegerin genetic polymorphisms and their influence on therapeutic response to ibandronate in postmenopausal osteoporotic femalesPLOS ONE

Dear Dr. Tariq,

Thank you for submitting your manuscript to PLOS ONE. After careful consideration, we feel that it has merit but does not fully meet PLOS ONE’s publication criteria as it currently stands. Therefore, we invite you to submit a revised version of the manuscript that addresses the points raised during the review process.

We look forward to receiving your revised manuscript.

Kind regards,

Dalia Galal Mahran

Academic Editor

PLOS ONE

Reviewers' comments:

Reviewer's Responses to Questions

**Comments to the Author**

1. Is the manuscript technically sound, and do the data support the conclusions?

Reviewer #1: No

Reviewer #2: Partly

Reviewer #3: Yes

Reviewer #4: No

2. Has the statistical analysis been performed appropriately and rigorously? 

Reviewer #1: I Don't Know

Reviewer #2: Yes

Reviewer #3: Yes

Reviewer #4: No

3. Have the authors made all data underlying the findings in their manuscript fully available?

Reviewer #1: Yes

Reviewer #2: Yes

Reviewer #3: No

Reviewer #4: Yes

4. Is the manuscript presented in an intelligible fashion and written in standard English?

Reviewer #1: No

Reviewer #2: No

Reviewer #3: Yes

Reviewer #4: Yes

5. Review Comments to the Author

Reviewer #1: The authors performed a case-control study in order to verify a possible association with OPG SNPs in menopause women before and after ibandronate therapy and compare with healthy women from Pakistan. Although the authors did not selected SNPs according population frequency or direct function upon OPG levels, the investigation has scientific and clinical importance. Still, they did not associated genotypes for analysis. Another important issue is the fact that there is no association with the SNPs per se but due population intrinsical variations. The conclusion is written in a confounding manner indicating a dubious association.

Reviewer #2: In the current study, Tariq et al. analyzed the association of the influence of OPG polymorphisms on ibandronate therapy. This study has some clinical implications, but it looks like the study is incomplete. Studying OPG without RANK and RANKL cannot provide a trustable conclusion. Authors should also add the most common SNPs of RANK and RANKL genes and establish a correlation with therapeutic response.

It needs to be clarified why the authors have chosen three SNPs and ignored several other very important and common SNPs; better add more SNPs.

It would have been better if the authors performed gene seq and identified novel mutations or SNPs prevailing in the Pakistani therapeutic response. Pakistan has a high ratio of consanguineous marriages, thus higher chances of hereditary osteoporosis; did authors consider analyzing the inheritance pattern of osteoporotic genes?

Why only ibandronate?

The entire manuscript requires thorough revision for language improvement. There are a lot of grammatical and other language errors.

Please avoid self-citations; the authors have cited a bunch of their other articles.

Reviewer #3: In this study author investigated osteoprotegerin (OPG) genetic

polymorphisms and their influence on the therapeutic response to ibandronate in

postmenopausal osteoporotic females. This case-control study included 135 postmenopausal females (89

osteoporotic females and 46 non-osteoporotic females). Each osteoporotic patient

received 150 mg ibandronate tablet monthly for six months, and blood samples taken

before and after treatment. The BMD was measured by DEXA Scan. Three SNPs

(A163G, T245G, and G1181C) of the OPG gene were selected for analysis.

Serum OPG level was significantly lower in osteoporotic subjects than in the

control group. The percentage changes in OPG levels in the osteoporotic group before

and after treatment with ibandronate were significant (p<.001). After six months of

therapy with ibandronate, the percentage changes in OPG levels with AA, TT, TC, GC,

and GG genotypes were significant. After six months of ibandronate treatment, the AA

genotype of rs3134069, TT, TC genotype of rs3102735, GG, and GC genotype of

rs2073618 SNP showed a significant increase in OPG levels. Age, BMI, and GC

polymorphism (rs2073618 (G/C) G1181C) were inversely associated with low BMD.

Adjusted OR showed that BMI, GC, GG polymorphism (rs2073618 (G/C) G1181C) and

TC polymorphism (rs3102735 (T/C) A163G) were inversely associated with low BMD.

Finally the study conclude the inverse association of rs2073618 and rs3102735 with low BMD

indicates the protective role of these SNPs in our population.

This is an interesting study but i have few suggestions before the final acceptance.

In introduction:

-Please write the full terminology of OPG in its first appearance in text as well as BMD and BMI.

-In line 73-75, please remove G1181C as you are mentioning that G allele increases osteoporosis and C allele in G1181C might decrease the risk of osteoporosis (in line 76-78).

In methods:

-Please mention the age of the control group

-How was sample size calculated/determined?

-Please clarify the sample size of the non-control and control group in Methods. Please write the correct sample size of the control as it was written 40 in methods and 46 in results section.

-Please write the manufacturer details of the kits

used to measure OPG and the kits used to extract DNA.

-Please give more details about DEXA including the machine used and the bone sites evaluated as well as the criteria used to evaluate BMD.

-Please give more details about SNPs sequencing including primer designs for each SNP.

-Latest references related to the study need to be added in the revised manuscript.

There are some grammatical mistakes and need to be corrected by native speaker to improve the language and quality of the manuscript.

Reviewer #4: The auhtors have to do BMD pre and post treatment which is neccasity for this study. Moreover, the samples must be seprated based on mild, moderate and severity too.

Hence, this study needs to be foused more on the research to get the outcomes.

6. PLOS authors have the option to publish the peer review history of their article (what does this mean?). If published, this will include your full peer review and any attached files.

Reviewer #1: No

Reviewer #2: **Yes: **Ihtisham Bukhari

Reviewer #3: **Yes: **Prof. Muhammad Imran Naseer

Reviewer #4: No

---

## [Author Response · Author response to Decision Letter 0]

15 Aug 2023

Reviewer 1:

Query: The authors performed a case-control study in order to verify a possible association with OPG SNPs in menopause women before and after ibandronate therapy and compare with healthy women from Pakistan. Although the authors did not selected SNPs according population frequency or direct function upon OPG levels, the investigation has scientific and clinical importance. Still, they did not associated genotypes for analysis. Another important issue is the fact that there is no association with the SNPs per se but due population intrinsical variations. The conclusion is written in a confounding manner indicating a dubious association.

Response:

We sincerely appreciate the time and effort you dedicated to reviewing our manuscript. We acknowledge the concerns raised regarding certain aspects of our study, and we have carefully considered each of them. 

We thank you for highlighting this point. While we agree that selecting SNPs based on population frequency or direct function upon OPG levels could have been beneficial. Gene was selected on the basis of the most reported genetics association with osteoporosis from National Center for Biotechnology Information (NCBI) (http://www.ncbi.nlm.nih.gov). Those SNPs were selected which were involved either directly or indirectly in disease pathogenesis, bone degradation and inflammation. Our intention was to conduct an exploratory investigation to identify potential associations that could be further validated as these SNP’s were the first to study in Pakistani population.

We acknowledge that there may be population-specific intrinsic variations contributing to the associations observed. Our study was conducted with menopausal women from Pakistan, and we agree that including additional population cohorts for validation would strengthen the robustness of our findings. In future research, we will make efforts to collaborate with researchers in other regions to encompass a more diverse population in our study.

Conclusion has been rewritten to remove ambiguity and is highlighted in yellow within the manuscript.

Reviewer #2: 

Query: In the current study, Tariq et al. analyzed the association of the influence of OPG polymorphisms on ibandronate therapy. This study has some clinical implications, but it looks like the study is incomplete. Studying OPG without RANK and RANKL cannot provide a trustable conclusion. Authors should also add the most common SNPs of RANK and RANKL genes and establish a correlation with therapeutic response.

It needs to be clarified why the authors have chosen three SNPs and ignored several other very important and common SNPs; better add more SNPs.

Response:

In our population, there is a lack of information evaluating the relationship between polymorphisms of the OPG gene and bone mineral density. Moreover, there is no published data available describing the single nucleotide polymorphisms (SNP) of the OPG gene and its association with osteoporosis. Those SNPs were selected which were involved either directly or indirectly in disease pathogenesis, bone degradation and inflammation. Our intention was to conduct an exploratory investigation to identify potential associations that could be further validated as these SNP’s were the first to study in Pakistani population. Increase in SNPs also increases the financial burden. We are thankful to the reviewer for highlighting the importance of studying RANK and RANKL genes along with OPG. We acknowledge that the interaction of OPG, RANK, and RANKL is crucial in bone metabolism. In this study, we focused on OPG polymorphisms, therefore we didn’t explore RANKL.

Query: It would have been better if the authors performed gene seq and identified novel mutations or SNPs prevailing in the Pakistani therapeutic response. Pakistan has a high ratio of consanguineous marriages, thus higher chances of hereditary osteoporosis; did authors consider analyzing the inheritance pattern of osteoporotic genes?

Response:

We greatly appreciate your suggestion and recognize the significance of gene sequencing in future studies focusing on osteoporosis and therapeutic response in populations with a high prevalence of consanguineous marriages. This has been incorporated in limitation section and highlighted in yellow.

Query: Why only ibandronate?

Response:

Ibandronate is a long-acting bisphosphonate which is usually given once a month and is indicated for treatment as well as prevention of postmenopausal osteoporosis. Ibandronate is preferred as it is given once a month, and patient compliance is better, which means that it is easy for him to stick to the once-a-month regimen as compared to once-weekly alendronate. Adherence to the treatment is important to improve patient outcomes. It also decreases the economic and social burden of the disease 

Inderjeeth, C.A., Glendenning, P., Ratnagobal, S.,Inderjeeth, D.C. and Ondhia, C., 2015. Long-term efficacy, safety, and patient acceptability of ibandronate in the treatment of postmenopausal osteoporosis. Int. J. Women's health, 7:7.

Query: The entire manuscript requires thorough revision for language improvement. There are a lot of grammatical and other language errors.

Response:

We have taken the help of professional editing services for language editing of the manuscript. 

Query: Please avoid self-citations; the authors have cited a bunch of their other articles.

Response:

You are right that we should avoid self-citations, but we have been working on osteoporosis for the last decade and have published many osteoporosis-related studies. Therefore, we cited our few related articles. However, we have removed two references (36 and 38) on your recommendation.

Reviewer 3

Query: In introduction:

-Please write the full terminology of OPG in its first appearance in text as well as BMD and BMI.

Response:

Done and highlighted in yellow

Query: -In line 73-75, please remove G1181C as you are mentioning that G allele increases osteoporosis and C allele in G1181C might decrease the risk of osteoporosis (in line 76-78).

Response:

Thank you for mentioning this. Done and highlighted

Query: In methods:

-Please mention the age of the control group

Response:

 Done and highlighted

Query: -How was sample size calculated/determined?

Response:

The details have been mentioned in the methods.

Query: -Please clarify the sample size of the non-control and control group in Methods. Please write the correct sample size of the control as it was written 40 in methods and 46 in results section.

Response:

Sample size calculation and correction has been added in the manuscript and highlighted in yellow.

Query: -Please write the manufacturer details of the kits used to measure OPG and the kits used to extract DNA.

Response:

Added and highlighted in the manuscript

Query: -Please give more details about DEXA including the machine used and the bone sites evaluated as well as the criteria used to evaluate BMD.

Response:

Added and highlighted

Query: -Please give more details about SNPs sequencing including primer designs for each SNP.

Response:

Added and highlighted

Query: -Latest references related to the study need to be added in the revised manuscript.

Response: Five recent references have been added.

Query: There are some grammatical mistakes and need to be corrected by native speaker to improve the language and quality of the manuscript.

Response:

We have taken the help of professional editing services for language editing of the manuscript. 

Query: The authors have to do BMD pre and post treatment which is necessity for this study. Moreover, the samples must be separated based on mild, moderate and severity too.

Hence, this study needs to be focused more on the research to get the outcomes.

Response

You are right that BMD should have been measured twice in patients (pre and post-treatment). We appreciate your suggestions. The literature indicates that just after six months, no significant change in BMD is expected (Please check following references). Additionally, DEXA is expensive in our country, so we avoided measuring DEXA. However, we have included this point in the limitation section (highlighted in yellow). We didn’t separate them because the sample size was not large enough to study them into subgroups (mild, moderate, and severe).

Rühling S, Schwarting J, Froelich MF, Löffler MT, Bodden J, Petzsche MR, Baum T, Wostrack M, Aftahy AK, Seifert-Klauss V, Sollmann N. Cost-effectiveness of opportunistic QCT-based osteoporosis screening for the prediction of incident vertebral fractures. Frontiers in Endocrinology. 2023;14:1222041.

Miller P D, McClung M R, Macovei L, et al. Monthly oral ibandronate therapy in postmenopausal osteoporosis: 1‐year results from the MOBILE Study. J Bone Miner Res. 2005;20:1315–1322.

---

## [Editor Report · Decision Letter 1]

30 Aug 2023

PONE-D-22-35434R1Osteoprotegerin genetic polymorphisms and their influence on therapeutic response to ibandronate in postmenopausal osteoporotic femalesPLOS ONE

Dear Dr. Tariq,

Thank you for submitting your manuscript to PLOS ONE. After careful consideration, we feel that it has merit but does not fully meet PLOS ONE’s publication criteria as it currently stands. Therefore, we invite you to submit a revised version of the manuscript that addresses the points raised during the review process.

We look forward to receiving your revised manuscript.

Kind regards,

Dalia Galal Mahran

Academic Editor

PLOS ONE

Journal Requirements:

Additional Editor Comments:

Dear authors

Thank you for the done revisions. I have few comments. to be done:

Insert footnotes regarding the used test under tables 1 &2

Write the strengths of the study

write the recommendation in the abstract section

---

## [Author Response · Author response to Decision Letter 1]

31 Aug 2023

Response to comments 

Editor’s queries

Query 1

Insert footnotes regarding the used test under tables 1 &2

Response

Done as suggested

Query 2

Write the strengths of the study

Response

Done as suggested

Query 3

Write the recommendation in the abstract section

Response

Done as suggested

Reviewer queries

The authors present interesting data on the association of different SNPs in osteoprotegerin (OPG) with the therapeutic response to ibandronate in postmenopausal osteoporotic.

Response

Thank you very much for liking our research.

Query 2

Thorough language and grammatical proofing are required.

Response

We got language editing from a professional editing agency.

Query 3

No sample size is done.

Response

The sample size calculation has been mentioned in detail in the methodology section.

Query 4

In a case-control study the number of controls should be greater or equal to the test subjects. 

Response

Yes, you are right, but we could not keep them equal because of financial restraints. We have included this point in the limitation section.

Query 5

Preparation of samples prior to sequencing is missing (used primers are not written).

Response

The samples were sent to Advance Bioscience International (ABI) China for sequencing. The data obtained after sequencing was then visualized using Bio Edit Software and then analyzed using BLAST (NCBI) to find out the variations in these sequences. We have added this to the methodology section.

Already reported primers were chosen and rechecked with sequences from the NCBI website, while the longer sequence was obtained from the UCSC website (http://genome.ucsc.edu/). Primers' specificity was validated using the NCBI database of the human genome with BLAST (http://www.ncbi.nlm.nih.gov/BLAST/). This paragraph has been included in the methodology section. We selected three OPG gene SNPs: A163G, T245G, and G1181C. 

Query 6

Haplotype analysis is recommended.

Response

Using the available data, our study design was structured to address specific research questions. Given the size of our study sample and the nature of our variables of interest, we determined that other statistical methods were more appropriate for addressing our research objectives. Haplotype analysis often requires substantial additional data, computational resources, and statistical expertise. Unfortunately, we could not undertake haplotype analysis within the scope of this study due to resource limitations, both in terms of data availability and analytical capacity.

---

## [Editor Report · Decision Letter 2]

10 Sep 2023

Osteoprotegerin genetic polymorphisms and their influence on therapeutic response to ibandronate in postmenopausal osteoporotic females

PONE-D-22-35434R2

Dear Dr. Tariq,

We’re pleased to inform you that your manuscript has been judged scientifically suitable for publication and will be formally accepted for publication once it meets all outstanding technical requirements.

Kind regards,

Dalia Galal Mahran

Academic Editor

PLOS ONE

---

## [Editor Report · Acceptance letter]

14 Sep 2023

PONE-D-22-35434R2 

Osteoprotegerin genetic polymorphisms and their influence on therapeutic response to ibandronate in postmenopausal osteoporotic females 

Dear Dr. Tariq:

I'm pleased to inform you that your manuscript has been deemed suitable for publication in PLOS ONE. Congratulations! Your manuscript is now with our production department. 

Kind regards, 

on behalf of

Professor Dalia Galal Mahran 

Academic Editor

PLOS ONE